# JAK-STAT pathway activation compromises nephrocyte function in a *Drosophila* high-fat diet model of chronic kidney disease

Yunpo Zhao[1,2†], Jianli Duan[1,2†], Hannah Seah[1,2], Joyce van de Leemput[1,2], Zhe Han[1,2]*

[1]Center for Precision Disease Modeling, Department of Medicine, University of Maryland School of Medicine, Baltimore, United States; [2]Division of Endocrinology, Diabetes and Nutrition, Department of Medicine, University of Maryland School of Medicine, Baltimore, United States

## eLife Assessment

This study presents **important** new insights linking obesity to kidney disease using a Drosophila model. A series of **compelling** experiments demonstrate that a high-fat diet induces excretion of a leptin-like JAK-STAT ligand from fat body, driving the adipose-nephrocyte axis through activated JAK-STAT signaling and subsequently causing a functional defect in nephrocytes. The approach using combination of genetic tools and pharmacological intervention is **solid** and confirms the mechanistic link, together with phenotypic analysis that further supports the authors conclusions.

**Abstract** Chronic kidney disease is a major health issue and is gaining prevalence. Using a *Drosophila* model for chronic kidney disease, we show that a high-fat diet (HFD) disrupts the slit diaphragm filtration structure in nephrocytes, the fly functional equivalent of mammalian podocytes. The structural disruption resulted in reduced filtration function in the affected nephrocytes. We demonstrate that HFD activates the JAK-STAT pathway in nephrocytes, which has previously been linked to diabetic kidney disease. JAK-STAT activation was initiated by increased expression and release of the adipokine, Upd2, from the fat body. This leptin-like hormone is a known ligand of JAK-STAT. Both genetic and pharmacological inhibition of JAK-STAT restored nephrocyte HFD-associated dysfunction. Altogether, our study reveals the importance of the JAK-STAT signaling pathway in the adipose tissue−nephrocyte axis and its contribution to HFD-associated nephropathy. These findings open new avenues for intervention in treating diabetic nephropathy and chronic kidney disease.

## Introduction

Chronic kidney disease is a prevalent health issue, with an estimated ~12% of people affected worldwide, many of whom are unaware of their condition (*Coresh, 2017*; *US Department of Health and Human Services and Centers for Disease Control and Prevention (Atlanta, GA), 2023*). The gradual loss of kidney function results in excess fluid and the buildup of metabolic waste compounds. Clinical symptoms include atherosclerosis, chronic inflammation, malnutrition, and insulin resistance among other metabolic imbalances (*Serrano et al., 2023*). Altogether, these lead to increased morbidity and mortality associated with chronic kidney failure (*Bikbov et al., 2020*). Diabetes, high blood

*For correspondence:
zhan@som.umaryland.edu

†These authors contributed equally to this work

**Competing interest:** The authors declare that no competing interests exist.

pressure, ageing, obesity, and increased BMI are major risk factors for glomerulopathy and chronic kidney disease (*Alizadeh et al., 2019*; *Berthoux et al., 2013*; *Bonnet et al., 2001*; *Coresh, 2017*; *Hsu et al., 2006*; *Moorhead et al., 1982*; *Tsuboi et al., 2013*). Notably, patients with primary kidney disease who are also obese have worsened outcomes (*Berthoux et al., 2013*), and following kidney transplantation, patients with increased BMI are at greater risk of adverse outcomes (*Curran et al., 2014*), with increasing risk as BMI increases (*Hsu et al., 2006*). The link between dietary fat intake and kidney disease has raised interest in the adipose-renal axis; that is, how do bodily fat deposits affect kidney function?

Podocytes from patients with chronic kidney disease contain lipid droplets that store excess fat (*Herman-Edelstein et al., 2014*; *Kimmelstiel and Wilson, 1936*). These cause lipotoxicity by disrupting the mitochondria, as well as endocytosis (*Lubojemska et al., 2021*), a process crucial to the kidney filtration structure (*Wang et al., 2021*). Studies in rats and mice on a HFD have repeatedly shown that the animals suffer from obesity, diabetes (altered insulin homeostasis), and kidney injury marked by functional (albuminuria; blood accumulation of BUN and creatinine; increased urinary biomarkers of kidney damage) and structural (glomerulopathy with glomerular hypertrophy and focal segmental glomerulosclerosis; fibrosis) deficiencies (*Altunkaynak et al., 2008*; *Ha et al., 2022*; *Jiang et al., 2005*; *Kuwahara et al., 2016*; *Lu et al., 2003*; *Rangel et al., 2019*; *Ruggiero et al., 2011*; *Sánchez-Navarro et al., 2021*; *Sun et al., 2020*; *Szeto et al., 2016*; *van der Heijden et al., 2015*). Like in patients with chronic kidney disease, lipid droplets have been observed in the kidneys of HFD rats and mice (*Deji et al., 2009*; *Jiang et al., 2005*; *Sun et al., 2020*; *van der Heijden et al., 2015*) and have been associated with dysfunctional cellular systems, including oxidative stress, renal inflammation, ER stress, disruption of mitochondrial dynamics, and impaired autophagy-lysosomal pathway in the cells of the kidneys (*Cai et al., 2024*; *Ha et al., 2022*; *Kuwahara et al., 2016*; *Li et al., 2016b*; *Lu et al., 2003*; *Rangel et al., 2019*; *Ruggiero et al., 2011*; *Sánchez-Navarro et al., 2021*; *Sun et al., 2020*; *Szeto et al., 2016*; *van der Heijden et al., 2015*). Altogether, these indicate the involvement of both local and systemic changes (*Deji et al., 2009*). However, our understanding of the pathways that govern the pathogenic effect of superfluous fat intake on kidney function, as well as how to leverage this knowledge to develop effective therapeutics, remains incomplete.

Recently, an HFD model in *Drosophila* showed lipotoxicity, i.e., lipid droplet accumulation, in the fly nephrocytes (*Lubojemska et al., 2021*). Nephrocytes share many characteristics with human podocytes, including genetics, molecular pathways, and function (*Weavers et al., 2009*; *Zhang et al., 2013a*; *Zhang et al., 2013b*). Both cell types contain highly specialized filtration structures known as slit diaphragms that act in concert with the basement membrane; the fly lacuna channel is similar to the urinary space in the mammalian Bowman's capsule; and, many key proteins for podocyte function are likewise essential for nephrocyte function (*van de Leemput et al., 2022*; *Wang et al., 2021*; *Weavers et al., 2009*; *Zhuang et al., 2009*). Indeed, fly in vivo nephrocyte models have been successfully used to study a variety of human kidney diseases, including forms of monogenic nephrotic syndrome and steroid-resistant nephrotic syndrome (SRNS) (*Ashraf et al., 2013*; *Bierzynska et al., 2022*; *Fu et al., 2017*; *Gee et al., 2015*; *Gee et al., 2013*; *Gonçalves et al., 2018*; *Hermle et al., 2018*; *Hermle et al., 2017*; *Milosavljevic et al., 2022*; *Odenthal et al., 2023*; *Paul et al., 2023*; *Zhao et al., 2019*; *Zhu et al., 2017*). The recent study using the HFD *Drosophila* model recapitulated the ectopic lipid droplets and cellular dysfunction observed in chronic kidney disease and found that the adipose-derived triglyceride lipase protects nephrocyte endocytosis under HFD conditions (*Lubojemska et al., 2021*). Here, we used an HFD *Drosophila melanogaster* model of chronic kidney disease to investigate the role and factors of the adipose-renal axis in chronic kidney disease and identified Upd-activated JAK-STAT signaling as a key component.

## Results

### HFD compromises nephrocyte function

To investigate the effect of superfluous fat consumption on nephrocyte function, newly eclosed flies were fed a normal fat diet (NFD) or a HFD for 7 days. Then, the flies were subjected to fluorescent dye uptake assays (*Weavers et al., 2009*; *Zhao et al., 2025*) to determine pericardial nephrocyte function. HFD leads to enlarged crop (*Liao et al., 2021*; *Zhao et al., 2023a*), which was used as an indication of food consistency. Compared to the NFD-fed flies, the HFD-fed flies showed reduced

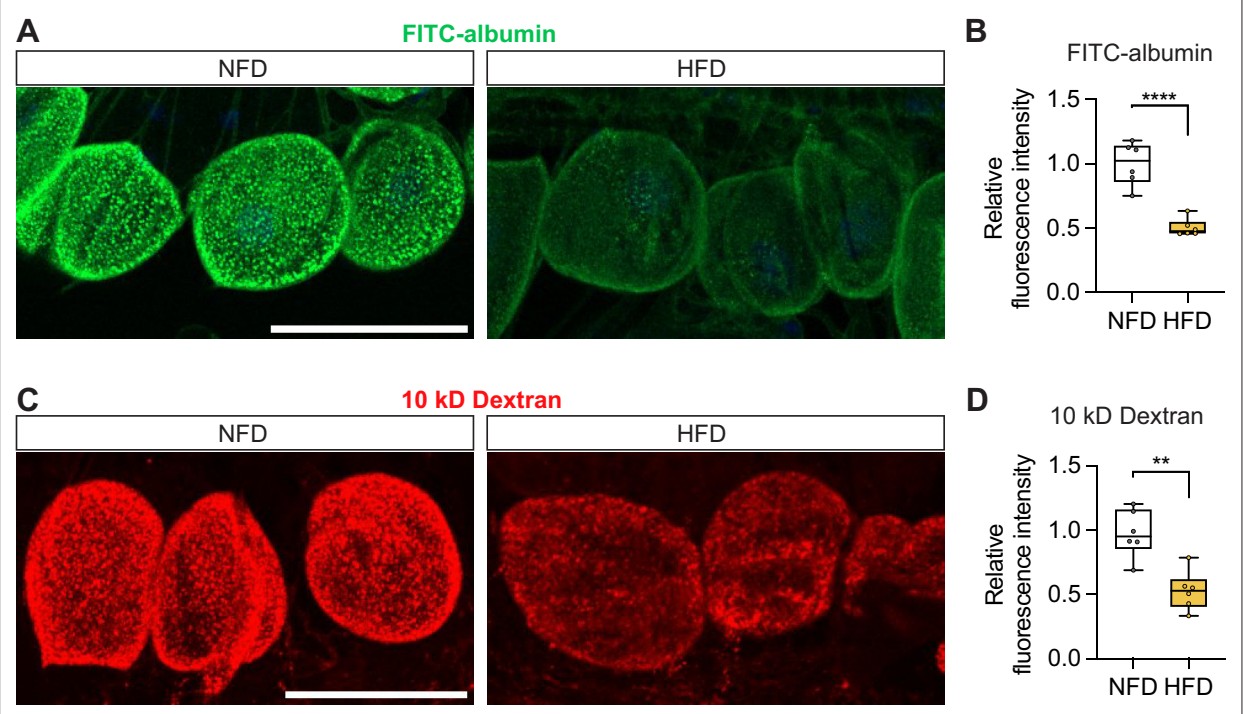

**Figure 1.** High-fat diet compromises nephrocyte function. Nephrocytes from control *Drosophila* (*w*[1118], females) fed a regular diet (normal fat diet, NFD) or high-fat diet (NFD supplemented with 14% coconut oil, HFD) for 7 days from eclosion. (**A**) Representative confocal images of nephrocytes show green fluorescence indicative of FITC-albumin uptake. Scale bar: 50 µm. (**B**) Box plot shows the quantitation of the relative fluorescence intensity of FITC-albumin shown in (**A**); middle line depicts the median and whiskers show minimum to maximum. Statistical analysis was performed with a two-tailed Student's t-test; ****$p<0.0001$; n=6 flies. (**C**) Representative confocal images of *Drosophila* nephrocytes (*w*[1118], 7 day-old females) show red fluorescence indicative of 10 kD dextran uptake. Scale bar: 50 µm. (**D**) Box plot shows the quantitation of the relative fluorescence intensity of 10 kD dextran shown in (**C**); middle line depicts the median and whiskers show minimum to maximum. Statistical analysis was performed with a two-tailed Student's t-test; **$p<0.01$; n=6 flies.

The online version of this article includes the following figure supplement(s) for figure 1:

**Figure supplement 1.** High-fat diet leads to lipid droplet accumulation in the nephrocytes.

FITC-conjugated albumin (FITC-albumin, 66 kD) fluorescence (*Figure 1A and B*). Likewise, HFD feeding significantly reduced 10 kD dextran uptake by the nephrocytes (*Figure 1C and D*). In line with HFD-caused lipid droplet accumulation in the larval nephrocytes (*Lubojemska et al., 2021*), we observed increased nephrocyte lipid droplets in the adults fed with HFD (*Figure 1—figure supplement 1*). Thus, confirming the HFD model in *Drosophila* for studying kidney disease (*Lubojemska et al., 2021*) and demonstrating that, in our hands, the consumption of superfluous fat caused nephrocyte uptake dysfunction.

## HFD alters nephrocyte morphology

The slit diaphragm (SD) is the fundamental filtration structure of nephrocytes. Thus, given the HFD-induced dysfunctional uptake, we next tested whether the SD is affected by HFD. Therefore, we looked at the distribution of polychaetoid (Pyd), the fly homolog of human tight junction protein 1 (TJP1, alias ZO-1) and a key component of the SD filtration unit (*van de Leemput et al., 2022*). Immunostaining with anti-Pyd antibody showed predominant membrane localization of Pyd in nephrocytes from NFD-fed flies. At the cortical surface of NFD nephrocytes, Pyd showed the fingerprint-like pattern characteristic of the SD (*Figure 2A*). However, in the HFD-fed flies, the SD fingerprint-like pattern appeared irregular and Pyd showed an uneven distribution with spots of increased signal intensity (*Figure 2A*). These might correlate to the trails of Pyd seemingly hanging from the cortical surface inside the HFD nephrocytes, along with cytosolic accumulation of Pyd protein (*Figure 2A and B*). Next, we used Sns-mRuby3, in which mRuby3 was tagged at the C-terminal of endogenous Sns (*Delaney et al., 2024*; *Zhao et al., 2025*), to verify the HFD effect on SD structure. We observed

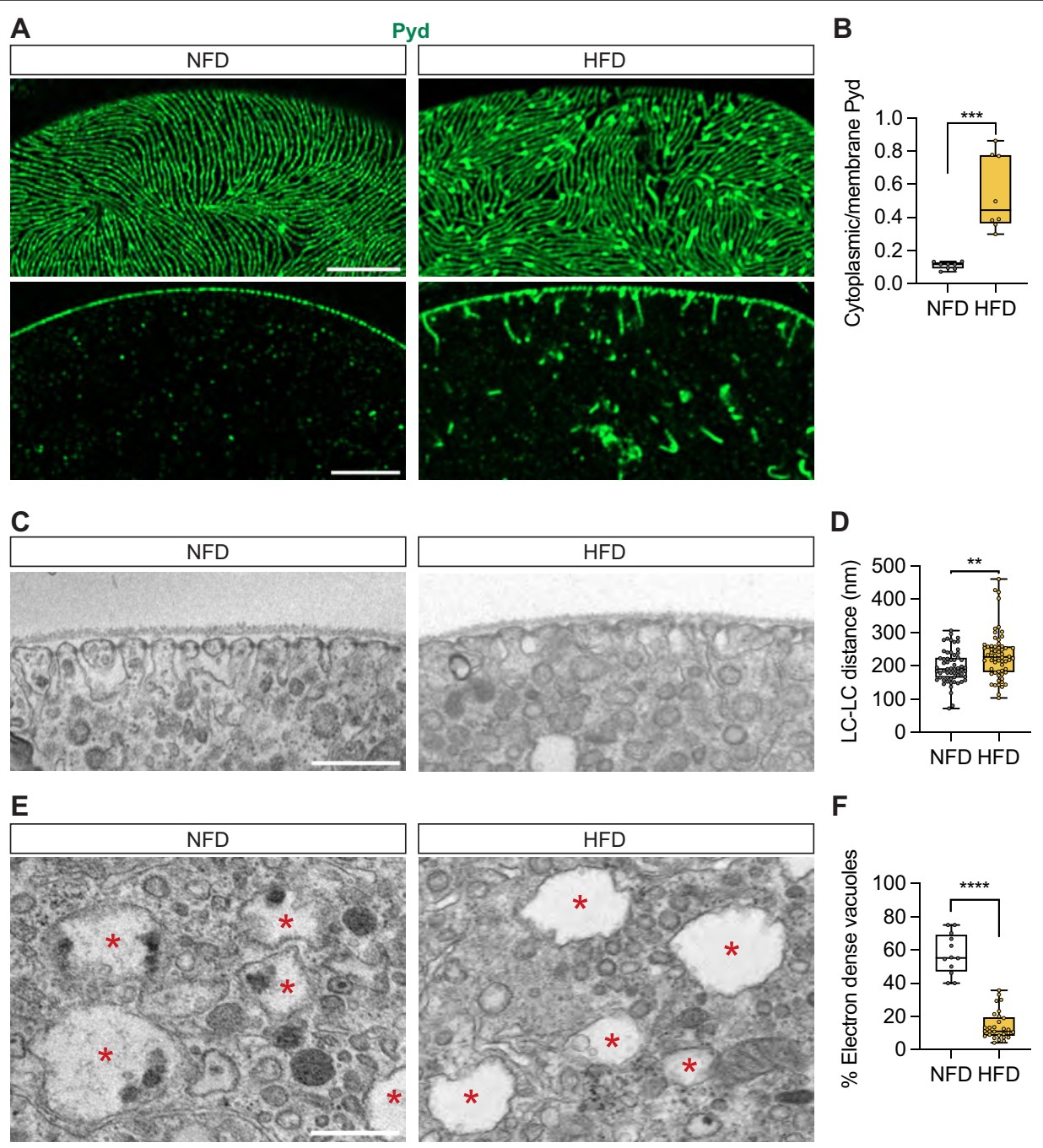

**Figure 2.** High-fat diet changes nephrocyte morphology. Nephrocytes from control *Drosophila* (*w*[1118], 7-day-old females) fed a regular diet (normal fat diet, NFD) or high-fat diet (NFD supplemented with 14% coconut oil, HFD). (**A**) Representative confocal images of *Drosophila* nephrocytes immunostained with anti-polychaetoid (Pyd) in green. Upper panels show cortical surface; Scale bar: 5 μm. Lower panels show subcortical regions; Scale bar: 5 μm. (**B**) Quantitation of Pyd protein distribution (cytoplasmic vs membrane); middle line depicts the median and whiskers show minimum to maximum. Statistical analysis was performed with a two-tailed Student's t-test; ***$p<0.001$; n=8 nephrocytes (1 nephrocyte/fly) from 7-day-old female flies. (**C**) Transmission electron microscopy (TEM) images of *Drosophila* nephrocyte (*w*[1118], 7-day-old females) cortical regions. Scale bar: 0.5 μm. (**D**) Quantitation of lacuna channel (LC)-LC distance based on images in (**C**); middle line depicts the median and whiskers show minimum to maximum. Statistical analysis was performed with a two-tailed Student's t-test; **$p<0.01$; n=60 LC-LC distance measurements obtained in 10 nephrocytes from six 7-day-old female flies for each group. (**E**) TEM images of *Drosophila* nephrocyte (*w*[1118], 7-day-old females) cytoplasmic regions. Red asterisks indicate large vacuoles. Scale bar: 0.5 μm. (**F**) Quantitation of the vacuoles that contain electron-dense structures based on images in (**E**). The middle line depicts the median and whiskers show minimum to maximum. Statistical analysis was performed with a two-tailed Student's t-test; ****$p<0.0001$; n=12 nephrocytes for NFD and 29 nephrocytes for HFD from six 7-day-old female flies.

*Figure 2 continued on next page*

*Figure 2 continued*

The online version of this article includes the following figure supplement(s) for figure 2:

**Figure supplement 1.** High-fat diet changes nephrocyte morphology.

cytoplasm retention of Sns-mRuby3 in the nephrocytes of HFD-fed flies (*Figure 2—figure supplement 1*). These findings show that a HFD leads to structural disruption of the SD filtration unit in nephrocytes, that likely accounts for the nephrocyte functional deficit observed earlier.

To study the nephrocyte SD and cytoplasmic regions in more detail, we performed transmission electron microscopy (TEM). The nephrocytes of NFD-fed flies showed regularly arranged lacuna channels along the cortical surface (*Figure 2C*). The distance between the lacuna channels was significantly increased in nephrocytes from HFD-fed flies (*Figure 2C and D*). Nephrocytes from both NFD and HFD fed flies showed big subcellular vacuoles (*Figure 2E*). However, the vacuoles in NFD nephrocytes frequently contained electron-dense structures, whereas most vacuoles in HFD nephrocytes were clear (*Figure 2E and F*). Altogether, these findings demonstrate that HFD causes significant changes to the nephrocyte and its SD, both at the cortical and subcortical levels.

## HFD potentiates the JAK-STAT pathway in nephrocytes

Activity of the Janus kinase/signal transducer and activator of transcription (JAK-STAT) pathway has been linked to diabetic kidney disease (*Berthier et al., 2009*); likewise, it is activated systemically in *Drosophila* in response to a chronic lipid-rich diet (*Woodcock et al., 2015*). Key components of the JAK-STAT pathway are conserved in flies, including *Signal-transducer and activator of transcription protein at 92E* (*Stat92E*), *hopscotch* (*hop*), *domeless* (*dome*), and *Suppressor of cytokine signaling at*

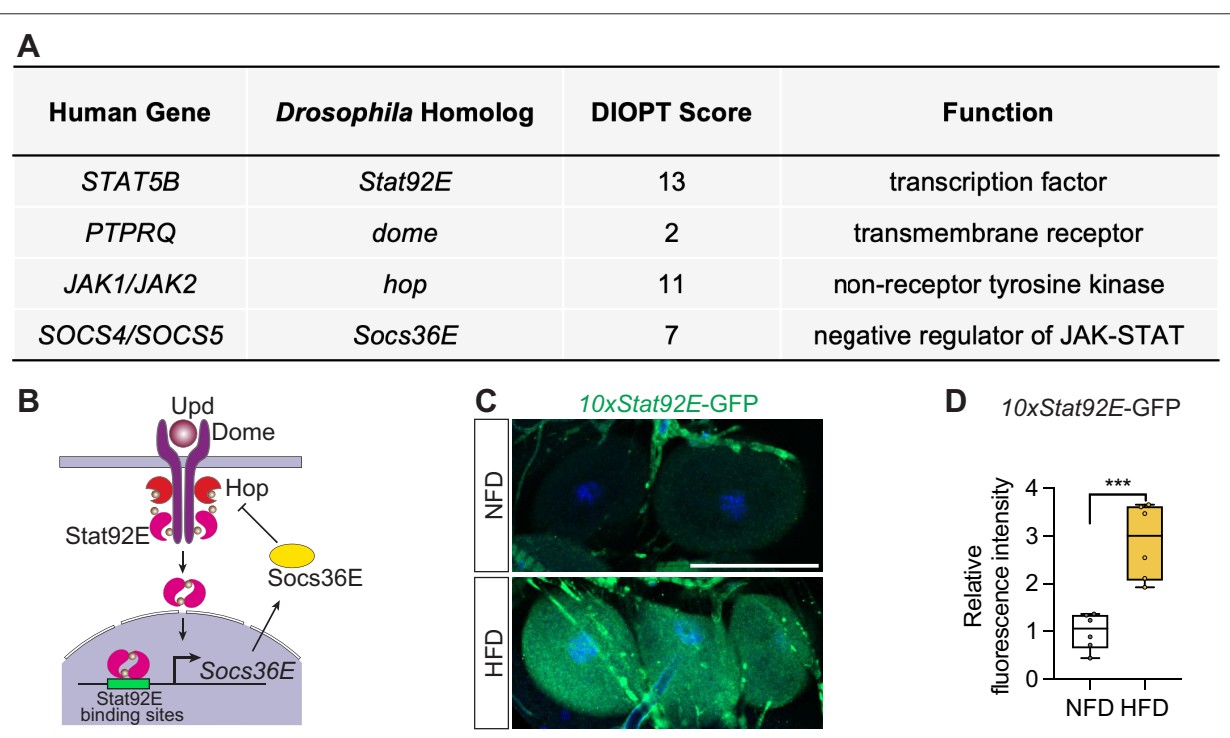

**A**

| Human Gene | *Drosophila* Homolog | DIOPT Score | Function |
|---|---|---|---|
| *STAT5B* | *Stat92E* | 13 | transcription factor |
| *PTPRQ* | *dome* | 2 | transmembrane receptor |
| *JAK1/JAK2* | *hop* | 11 | non-receptor tyrosine kinase |
| *SOCS4/SOCS5* | *Socs36E* | 7 | negative regulator of JAK-STAT |

**Figure 3.** High-fat diet activates the Janus kinase/signal transducer and activator of transcription (JAK-STAT) pathway in nephrocytes. (**A**) Table lists human genes encoding JAK-STAT pathway components, along with their *Drosophila* homologs, the DRSC Integrative Ortholog Prediction Tool (DIOPT) score (maximum score = 15), and their function. (**B**) Graphical representation of the JAK-STAT signaling pathway and interaction between its components. Domeless, Dome; JAK Hopscotch, Hop; Signal-transducer and activator of transcription 92E, Stat92E; Suppressor of cytokine signaling at 36E, Socs36E; Unpaired, Upd. (**C**) Representative confocal images of nephrocytes from control *Drosophila* (*10xStat92E*-GFP, 7-day-old females) fed a regular diet (normal fat diet, NFD) or high-fat diet (HFD, NFD supplemented with 14% coconut oil). *10xStat92E*-GFP is shown in green fluorescence; DAPI (blue) stains DNA to visualize the nucleus. Scale bar: 50 µm. (**D**) Box plot shows the quantitation of the relative fluorescence intensity of *10xStat92E*-GFP based on images in (**C**); middle line depicts the median and whiskers show minimum to maximum. Statistical analysis was performed with a two-tailed Student's t-test; ***p<0.001; n=6 flies.

*36E* (*Socs36E*) (*Figure 3A and B*). Therefore, we tested whether the JAK-STAT pathway was activated in the nephrocytes of HFD-fed flies. We used a fluorescent reporter *10xStat92E*-GFP of JAK-STAT pathway activity (*Ekas et al., 2006*). Low levels of *Stat92E*-GFP fluorescence were detected in the nephrocytes of flies on a regular diet (NFD). However, in nephrocytes from HFD-fed flies, the fluorescence was significantly increased (*Figure 3C and D*). These findings indicate that the consumption of superfluous fat indeed activates the JAK-STAT pathway in nephrocytes.

## Activation of the Janus kinase, Hop, reduces nephrocyte function

Since JAK-STAT signaling has been shown to play an important role during development (*Brown et al., 2001*; *Liu et al., 2009*), we wanted to isolate its requirement for mature nephrocyte filtration function. To do this, we used a dominant gain-of-function allele of *hop* (*Tumorous-lethal*, *hop.Tum*); a single amino acid change (Gly341Glu) whose expression leads to JAK-STAT pathway activation (*Harrison et al., 1995*). We expressed *UAS-hop.Tum* specifically in mature nephrocytes using a temperature-sensitive *Dot*-Gal4 driver (*Dot*-Gal4; *tub*-Gal80ts, referred to as *Dot*-Gal4ts), known as the TARGET system (*McGuire et al., 2004*), to turn on expression in adult flies specifically as they are switched to a different environmental temperature (*Figure 4A*). Overexpression of *hop.Tum* significantly reduced nephrocyte absorption ability in the adults, as shown with reduced FITC-albumin and 10 kD dextran in *Dot*-Gal4ts >UAS *hop.Tum* nephrocytes compared with control nephrocytes (*Figure 4B–E*).

To validate these results and remove between-fly variability, both biological and technical, we also tested the effect of JAK-STAT pathway activation in tissue mosaic clones using Flp-out (*Delaney et al., 2024*; *Duan et al., 2020*; *Figure 4F*). For this technique, first instar larvae are exposed to heat shock to activate the Hsp70P transcription factor, which induces *Flippase* (*Flp*). Flp activity, in turn, removes the stop cassette, thereby driving Gal4 expression. Gal4 drives the expression of UAS-GFP and UAS-*hop.tum* within the same cell. Thus, *hop.Tum* is only expressed in GFP-labelled nephrocyte clones; whereas, GFP-negative nephrocytes are indicative of no Gal4 being expressed and serve as internal control cells (*Figure 4F*). In line with the TARGET assays, UAS-*hop.Tum* overexpression significantly reduced nephrocyte absorption ability in adult flies (*Figure 4G and H*). These data support a role for the JAK-STAT pathway in nephrocyte function.

## Depletion of Socs36E, a negative regulator of JAK-STAT, or increased adipokine Upd2, a JAK-STAT ligand, results in decreased nephrocyte function

Next, we looked at additional JAK-STAT pathway components in the context of nephrocyte function. Suppressor of cytokine signaling at 36E (Socs36E) is transcriptionally regulated by Signal-transducer and activator of transcription 92E (Stat92E) and functions as a negative regulator of JAK-STAT signaling (*Callus and Mathey-Prevot, 2002*; *Karsten et al., 2002*; *Figure 3B*), and it is expressed in the nephrocytes (*Figure 3A*). Silencing *Socs36E* specifically in nephrocytes (*Dot*-Gal4 >*Socs36E*-RNAi) significantly reduced nephrocyte uptake function as evident in significantly decreased FITC-albumin (*Figure 5A and B*) and 10 kD dextran (*Figure 5C and D*) fluorescence compared to the control nephrocytes.

In *Drosophila*, the JAK-STAT pathway ligand unpaired (Upd) family is encoded by *upd1*, *upd2*, and *upd3*. These ligands bind to Domeless (Dome), a single-transmembrane receptor, to activate JAK-STAT signaling (*Figure 3B*). Of these, Upd2 is functionally equivalent to human Leptin and is highly expressed in the fat body, the fly's functional equivalent of vertebrate adipose tissue (*Hombría et al., 2005*; *Rajan et al., 2017*; *Rajan and Perrimon, 2012*). In *Drosophila*, HFD has been shown to upregulate Upd2 expression and secretion (*Rajan et al., 2017*; *Rajan and Perrimon, 2012*). In our model, overexpression of Upd2, specifically in the fat body (*ppl*-Gal4 driver; *ppl*-Gal4>*upd2* GFP), significantly reduced 10 kD dextran uptake in the nephrocytes compared to the control (*ppl*-Gal4/+) (*Figure 5E and F*), indicating reduced nephrocyte function. Since our *upd2* over-expression line contains *upd2*-GFP, we could not use the FITC-albumin uptake assay, as both signals occupy the same imaging channel. Upd2-GFP signal was found in the nephrocytes in *ppl*-Gal4>*upd2* GFP flies (*Figure 5—figure supplement 1A*), indicating an inter-tissue communication of Upd2-GFP. Notably, Upd2-GFP overexpression in the fat body caused cytoplasm retention of Sns-mRuby3 (*Figure 5G and H*) and compromised the cortical fingerprint pattern (*Figure 5—figure supplement 1B*).

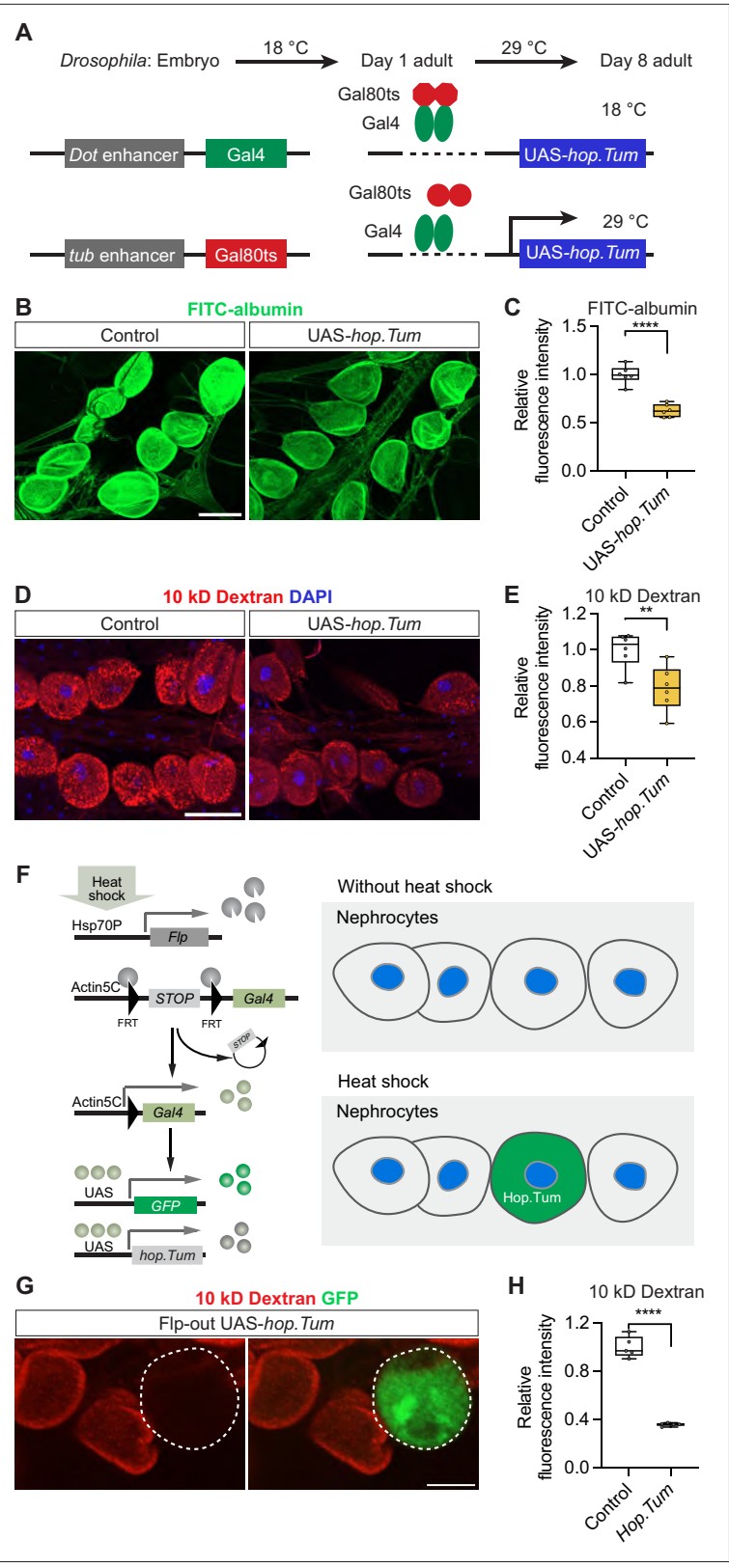

**Figure 4.** Janus kinase/signal transducer and activator of transcription (JAK-STAT) pathway activation compromises nephrocyte function. (**A**) Schematic illustration of targeted UAS-*hop.Tum* expression in the nephrocytes; *hopscotch.Tumorous-lethal*, dominant gain-of-function, constitutively activates JAK-STAT. Temperature-sensitive Gal80ts binds to Gal4 and acts as a negative regulator of the Gal4 transcriptional activator at 18°C. A temperature

*Figure 4 continued on next page*

*Figure 4 continued*

switch to 29°C releases Gal80ts inhibition as it can no longer bind Gal4, thus allowing UAS-*hop.Tum* expression driven by Gal4 to occur. A timeline for temperature switches of the fly at different stages of development have been indicated. (**B**) Representative confocal images of FITC-albumin fluorescence (green) in nephrocytes from control flies (*Dot*-Gal4/+; *tub*-Gal80ts/+) and those with activated JAK-STAT (*Dot*-Gal4/UAS-*hop.Tum*; *tub*-Gal80ts/+). Scale bar: 50 µm. (**C**) Box plot shows the quantitation of the relative fluorescence intensity of FITC-albumin based on images in (**B**); middle line depicts the median and whiskers show minimum to maximum. Statistical analysis was performed with a two-tailed Student's t-test; ****$p<0.0001$; n=6 flies (7-day-old females). (**D**) Representative confocal images of 10 kD dextran fluorescence (red) in nephrocytes from control flies (*Dot*-Gal4/+; *tub*-Gal80ts/+) and those with activated JAK-STAT (*Dot*-Gal4/UAS-*hop.Tum*; *tub*-Gal80ts/+); DAPI (blue) stains DNA to visualize the nucleus. Scale bar: 50 µm. (**E**) Box plot shows the quantitation of the relative fluorescence intensity of 10 kD dextran uptake based on images in (**D**); middle line depicts the median and whiskers show minimum to maximum. Statistical analysis was performed with a two-tailed Student's t-test; **$p<0.01$; n=6 flies (7-day-old females). (**F**) Schematic illustration of the Flippase (Flp)-out clone strategy to induce UAS-*hop.Tum* expression. Heat shock induces the expression of Flp recombinase, which excises a stop cassette to initiate Gal4 expression. Gal4 binding to the upstream activation sequences (UAS) drives the expression of GFP (as a marker for positive Flp-out clones) and UAS-*hop.Tum*. (**G**) Representative confocal images of 10 kD dextran fluorescence (red) in nephrocytes from flies with a GFP labeled Flp-out UAS-*hop.Tum* clone (*hs-Flp*[122]/+; UAS-*Flp*[JD1]/UAS-*hop.Tum*; *Act5C>CD2>Gal4*[S], UAS-*mCD8GFP*[LL6]/+). (**H**) Box plot shows the quantitation of the relative fluorescence intensity of 10 kD dextran fluorescence uptake based on images in (**G**); middle line depicts the median and whiskers show minimum to maximum. Control neighbor of Flp-out clone; UAS-*hop.Tum* (clone). Statistical analysis was performed with a two-tailed Student's t-test; ****$p<0.0001$; n=5 clones and five neighbor cells.

Like the Janus kinase Hop, the negative regulator Socs36E and the ligand adipokine Upd2 are additional JAK-STAT pathway components that are required for nephrocyte function.

## HFD-induced nephrocyte functional defects are mitigated by Stat92E-mediated inhibition of JAK-STAT

To determine that JAK-STAT forms a direct link between HFD and nephrocyte dysfunction, we knocked down *Stat92E* (*Stat92E*-IR; *Recasens-Alvarez et al., 2017*) in the nephrocytes to disrupt the JAK-STAT signaling pathway (*Figure 3B*). Deficiency for Stat92E did not affect the uptake function (FITC-albumin or 10 kD dextran) of adult nephrocytes in flies fed a NFD (*Figure 6A–D*). However, under HFD conditions, in which nephrocytes from control flies showed significantly reduced uptake, *Stat92E*-deficient nephrocytes showed FITC-albumin and 10 kD dextran fluorescence restored to NFD levels (*Figure 6A–D*). We used an independent *Stat92E-RNAi* line in the functional assays and obtained similar results, showing that dextran fluorescence was restored to NFD levels in *Stat92E*-deficient nephrocytes under HFD conditions (*Figure 6—figure supplement 1*). Interestingly, under HFD conditions, the cytoplasm retention of Sns-mRuby3 was restored to NFD levels in *Stat92E*-deficient nephrocytes (*Figure 6—figure supplement 2*). These findings demonstrate that Stat92E, and by extension the JAK-STAT pathway, is required for the nephrocyte dysfunction induced by HFD.

## HFD-induced nephrocyte functional defects can be attenuated by methotrexate treatment

Methotrexate suppresses STAT activation; it inhibits the phosphorylation of JAK which is necessary for JAK-STAT pathway activation and was shown to do so without affecting other phosphorylation-dependent pathways (*Thomas et al., 2015*). In our model system, methotrexate treatment (10 µM; 60 min) reduced levels of Stat92E (10xStat92E-GFP) in the nephrocytes (*Figure 7—figure supplement 1*). Stat92E is a key component of the pathway; its reduction is indicative of decreased JAK-STAT signaling. Next, we treated flies on NFD or HFD with 10 µM methotrexate (60 min incubation; ex vivo) to study the effect of pharmacological JAK-STAT inhibition on nephrocyte function. In nephrocytes from control flies on a regular diet (NFD), the methotrexate treatment had no effect on uptake of FITC-albumin or 10 kD dextran (*Figure 7A–D*). However, in flies on a HFD, treatment with methotrexate led to significantly increased nephrocyte FITC-albumin and 10 kD dextran uptake, restoring levels to those observed in control fly NFD nephrocytes (*Figure 7A–D*). We asked whether methotrexate treatment has an effect on Sns-mRuby3 subcellular distribution. In line with our prediction, under HFD conditions, pharmacological JAK-STAT inhibition restored Sns-mRuby3 distribution to NFD levels

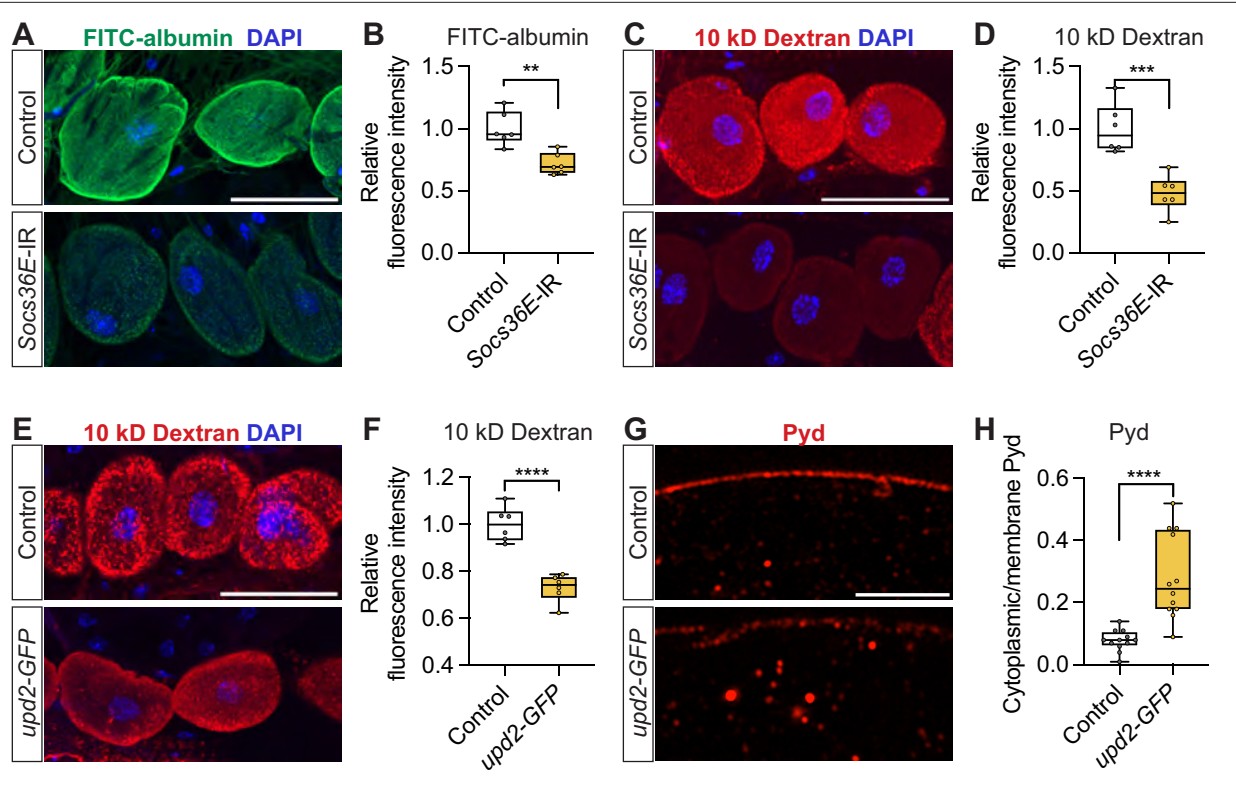

**Figure 5.** Silencing *Socs36E* in the nephrocytes, or *upd2* overexpression in the fat body, leads to nephrocyte dysfunction. (**A**) Representative confocal images of FITC-albumin (green) in nephrocytes from control flies (*Dot*-Gal4/+) and flies with nephrocyte-specific silencing of the Socs36E Janus kinase/signal transducer and activator of transcription (JAK-STAT) inhibitor (*Dot*-Gal4>*Socs36E*-IR); DAPI (blue) stains DNA to visualize the nucleus. Scale bar: 50 µm. *Socs36E, Suppressor of cytokine signaling at 36E*. (**B**) Box plot shows the quantitation of the relative fluorescence intensity of FITC-albumin uptake based on images in (**A**); middle line depicts the median and whiskers show minimum to maximum. Statistical analysis was performed with a two-tailed Student's t-test; **$p<0.01$; n=6 flies (7-day-old females). (**C**) Representative confocal images of 10 kD dextran fluorescence (red) in nephrocytes from control flies (*Dot*-Gal4/+) and flies with nephrocyte-specific silencing of the Socs36E JAK-STAT inhibitor (*Dot*-Gal4>*Socs36E*-IR); DAPI (blue) stains DNA to visualize the nucleus. Scale bar: 50 µm. *Socs36E, Suppressor of cytokine signaling at 36E*. (**D**) Box plot shows the quantitation of the relative fluorescence intensity of 10 kD dextran uptake based on images in (**C**); middle line depicts the median and whiskers show minimum to maximum. Statistical analysis was performed with a two-tailed Student's t-test; ***$p<0.001$; n=6 flies (7-day-old females). (**E**) Representative confocal images of 10 kD dextran fluorescence (red) in nephrocytes from control flies (*ppl*-Gal4/+) and flies with fat body-specific overexpression of JAK-STAT ligand Upd2 (*ppl*-Gal4>*upd2* GFP); DAPI (blue) stains DNA to visualize the nucleus. Scale bar: 50 µm. *ppl, pumpless; upd2, unpaired 2*. (**F**) Box plot shows the quantitation of the relative fluorescence intensity of 10 kD dextran uptake based on images in (**E**); middle line depicts the median and whiskers show minimum to maximum. Statistical analysis was performed with a two-tailed Student's t-test; ****$p<0.0001$; n=6 flies (7-day-old females). (**G**) Representative confocal images of nephrocytes from control flies (*ppl-Gal4/+*) and flies with fat body-specific overexpression of Upd2 (*ppl-Gal4*>*upd2* GFP). Anti-polychaetoid (Pyd) is shown in red. Scale bar: 4 µm. (**H**) Quantitation of Pyd protein distribution (cytoplasmic vs membrane); middle line depicts the median and whiskers show minimum to maximum. Statistical analysis was performed with a two-tailed Student's t-test; ****$p<0.0001$; n=12 nephrocytes (one nephrocyte/fly) from 7-day-old female flies.

The online version of this article includes the following figure supplement(s) for figure 5:

**Figure supplement 1.** Upd2-GFP is secreted from the fat body and transported to the nephrocytes.

---

(*Figure 7—figure supplement 2*). Thus, like the genetic intervention (*Stat92E* inhibition; *Figure 6*), pharmacological intervention with the JAK-STAT inhibitor methotrexate restored nephrocyte function caused by HFD. These findings support the notion that HFD-induced nephrocyte dysfunction is mediated by the JAK-STAT signaling pathway.

## Discussion

A recent publication showed that flies fed a HFD can recapitulate key features of chronic kidney disease, including lipid droplet formation, altered mitochondria dynamics, and endocytosis defects observed as reduced uptake of dextran and albumin (*Lubojemska et al., 2021*). This previous study

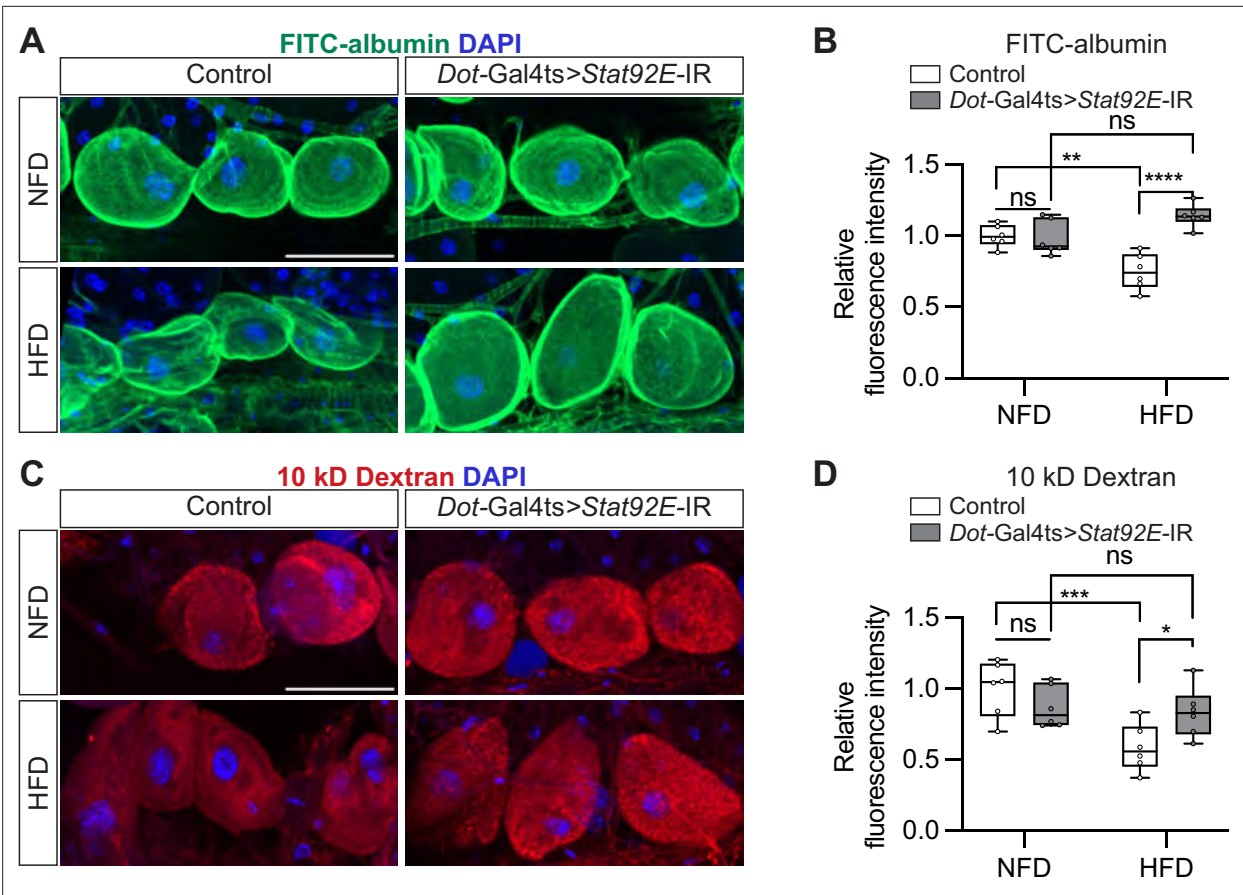

**Figure 6.** Silencing *Stat92E* attenuates nephrocyte functional defects caused by a high-fat diet. Nephrocytes from control flies (*Dot*-Gal4/+; *tub*-Gal80ts/+) and those with *Stat92E* silencing as adults (*Dot*-Gal4/UAS-*Stat92E*-IR; *tub*-Gal80ts/+). UAS-*Stat92E*-RNAi expression was induced at the adult stage (see *Figure 4A*) for seven days before the uptake assay. *Stat92E, Signal-transducer and activator of transcription 92E*. (**A**) Representative confocal images of FITC-albumin fluorescence (green); DAPI (blue) stains DNA to visualize the nucleus. Scale bar: 50 µm. (**B**) Box plot shows the quantitation of the relative fluorescence intensity of FITC-albumin uptake based on images in (**A**); middle line depicts the median and whiskers show minimum to maximum. Statistical analysis was performed by two-way ANOVA with Sidak correction; **$p<0.01$; ****$p<0.0001$; ns, not significant; n=6 flies (7-day-old females). (**C**) Representative confocal images of 10 kD dextran fluorescence (red); DAPI (blue) stains DNA to visualize the nucleus. Scale bar: 50 µm. (**D**) Box plot shows the quantitation of the relative fluorescence intensity of 10 kD dextran uptake based on images in (**C**); middle line depicts the median and whiskers show minimum to maximum. Statistical analysis was performed by two-way ANOVA with Sidak correction; ***$p<0.001$; ****$p<0.0001$; ns, not significant; n=6 flies (7-day-old females).

The online version of this article includes the following figure supplement(s) for figure 6:

**Figure supplement 1.** Stat92E depletion rescues HFD-caused nephrocyte functional decline.

**Figure supplement 2.** Stat92E depletion rescues HFD-caused Sns-mRuby3 distribution defects in the nephrocytes.

found that excess fatty acids, a sign of lipotoxicity, due to a HFD are released from adipose tissue into circulation, then filtered out by the nephrocytes by receptor-mediated endocytosis, at which point the fatty acids accumulate in the lipid droplets (*Lubojemska et al., 2021*), like those observed in the podocytes of patients with chronic kidney disease (*Herman-Edelstein et al., 2014*; *Kimmelstiel and Wilson, 1936*). Notably, lipid droplet lipolysis and PGC1α could counteract the HFD-induced disrupted endocytosis in the nephrocytes via the mitochondria (*Lubojemska et al., 2021*). This demonstrated one mechanism by which excess lipid droplets, due to HFD, can disrupt nephrocyte kidney function. Here, we likewise used a HFD *Drosophila* model to study chronic kidney disease and revealed another mechanism by which HFD affects both nephrocyte uptake function and morphology of its filtration structure, the slit diaphragm. HFD upregulates the expression of the adipokine Upd2, the functional homolog to human leptin, which activates the JAK-STAT pathway (*Rajan and Perrimon, 2012*; *Zhao et al., 2023b*). Of note, a previous study has shown that Stat1 exerts a repressive effect on PGC1α transcription (*Sisler et al., 2015*). These observations suggest a potential mechanistic

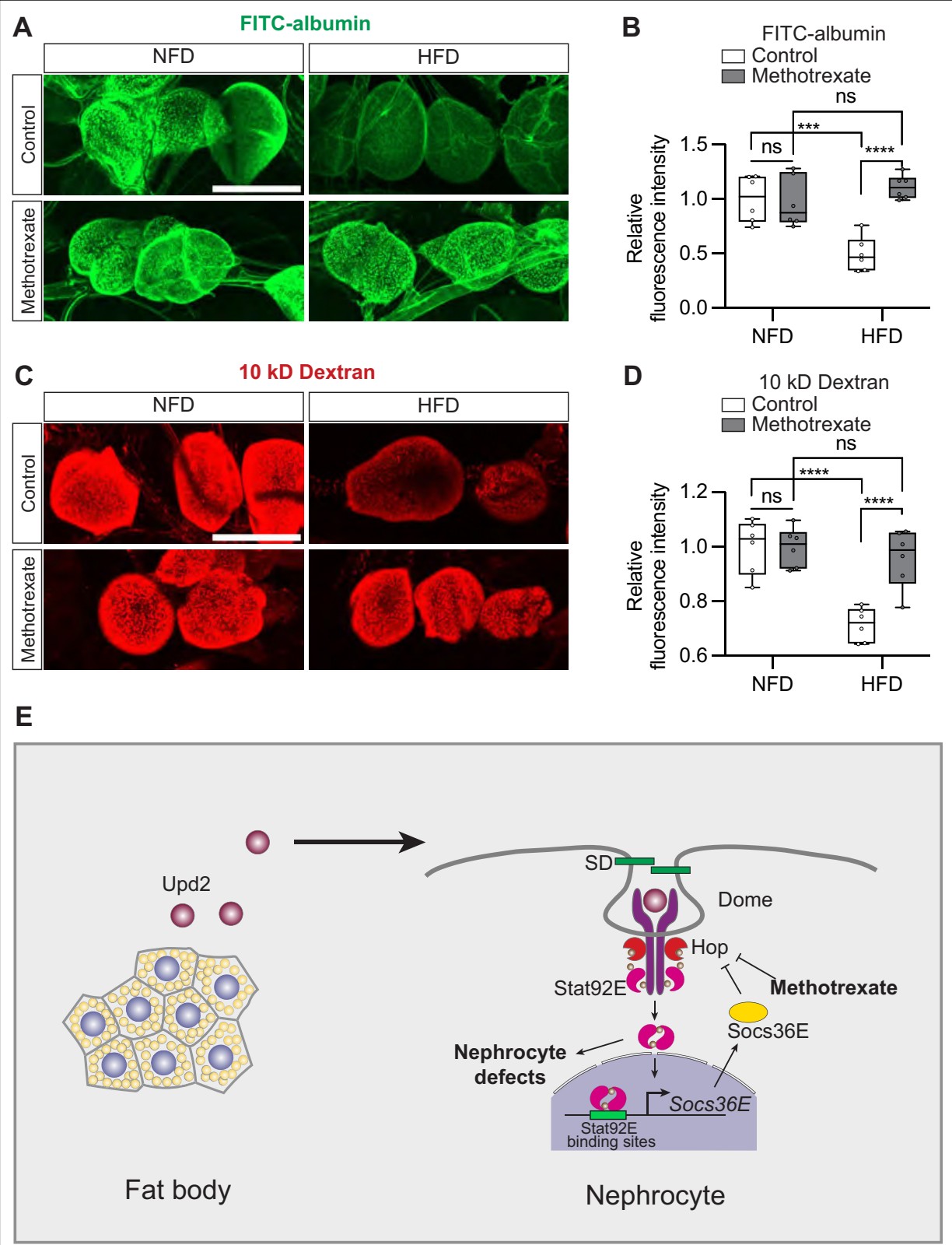

**Figure 7.** Methotrexate treatment can restore nephrocyte function following a high-fat diet. Nephrocytes from control *Drosophila* (*w*[1118]; 7-day-old females) fed a regular diet (normal fat diet, NFD) or high-fat diet (NFD supplemented with 14% coconut oil, HFD), with or without methotrexate (10 µM; ex vivo 60 min) treatment. (**A**) Representative confocal images of FITC-albumin fluorescence (green). Scale bar: 50 µm. (**B**) Box plot shows the quantitation of the relative fluorescence intensity of FITC-albumin uptake based on images in (**A**); middle line depicts the median and whiskers show

*Figure 7 continued on next page*

*Figure 7 continued*

minimum to maximum. Statistical analysis was performed by two-way ANOVA with Sidak correction; ***$p<0.001$, ****$p<0.0001$; ns, not significant; n=6 flies (7-day-old females). (**C**) Representative confocal images of 10 kD dextran fluorescence (red). Scale bar: 50 µm. (**D**) Box plot shows the quantitation of the relative fluorescence intensity of 10 kD dextran uptake based on images in (**C**); middle line depicts the median and whiskers show minimum to maximum. Statistical analysis was performed by two-way ANOVA with Sidak correction; ****$p<0.0001$; ns, not significant; n=6 flies (7-day-old females). (**E**) Graphic of proposed model for high-fat diet-induced nephrocyte defects via an adipose-nephrocyte axis. A high-fat diet upregulates the expression and secretion of the adipokine Unpaired 2 (Upd2), leptin-like hormone, from the fat body. Upd2 is a Janus kinase/signal transducer and activator of transcription (JAK-STAT) ligand, and it activates JAK-STAT signaling at the nephrocytes (Signal-transducer and activator of transcription 92E, Stat92E; Suppressor of cytokine signaling at 36E, Socs36E; JAK Hopscotch, Hop; Domeless, Dome). The overactive JAK-STAT pathway disrupts the integrity of the slit diaphragm (SD) filtration structure and thereby leads to nephrocyte dysfunction.

The online version of this article includes the following figure supplement(s) for figure 7:

**Figure supplement 1.** Methotrexate treatment inhibits Janus kinase/signal transducer and activator of transcription (JAK-STA)T pathway activity.

**Figure supplement 2.** Methotrexate treatment restores Sns-mRuby3 distribution defects following a high-fat diet.

interaction between Jak/Stat signaling and PGC1α regulation. Our data show this also holds true in the nephrocytes, and as such, compromises nephrocyte function (*Figure 7E*). These findings support obesity as a causal factor in chronic kidney disease.

Of note, in early-stage diabetic kidney disease, JAK-STAT pathway genes are upregulated in patient podocytes (*Berthier et al., 2009*). In a rat diabetic model, treatment with a JAK-STAT inhibitor (AG-490) reduced proteinuria (*Banes et al., 2004*); and overexpression of JAK2 in diabetic mouse podocytes elevated JAK-STAT pathway activity and exacerbated diabetic kidney disease (*Zhang et al., 2017*). These findings, like ours that showed effective treatment of HFD nephropathy using the JAK-STAT inhibitor (methotrexate), support the involvement of the JAK-STAT pathway in HFD-related chronic kidney disease. They also demonstrate that this pathological effect is conserved from flies to mammals. In fact, a phase 2 clinical trial demonstrated that the small molecule baricitinib, a selective JAK1 and JAK2 inhibitor, effectively lowered albuminuria in patients with type 2 diabetes and diabetic kidney disease (*Tuttle et al., 2018*). Altogether, these studies support JAK-STAT inhibition as a therapeutic intervention for nephropathy associated with superfluous fat intake (*Brosius et al., 2016*). The conserved disease mechanism makes the HFD fly model a valuable platform to screen JAK-STAT inhibitors for their efficacy to treat chronic kidney disease. The fly findings showed a direct JAK-STAT link at the adipose tissue–nephrocyte axis that leads to nephrocyte dysfunction, via HFD upregulated expression of the adipokine Upd2, which activates the JAK-STAT pathway (*Rajan and Perrimon, 2012*; *Figure 7E*). In support, like in our fly model, increased leptin has been observed in obese patients (*Considine et al., 1996*) and in patients with chronic renal failure (*Heimbürger et al., 1997*).

Finally, while growing evidence demonstrates that a pharmacological block of the JAK-STAT pathway could effectively treat nephropathy, for decades, the first-line treatment for obesity-related glomerulopathy has been RAS inhibitors (*Jiang et al., 2023*). Their beneficial effects stem from lowering blood pressure and by lowering the estimated glomerular filtration rate (eGFR) (*Banerjee et al., 2022*; *Yamout et al., 2014*). One such inhibitor, telmisartan, targets RAS by blocking the angiotensin (Ang) II receptor and was shown effective in treating nephropathy in a rat model of metabolic syndrome (HFD-fed rats) (*Li et al., 2016a*). Notably, telmisartan treatment decreased leptin release from adipose tissue, thereby supporting our model and implicating a hitherto unknown mechanism that contributes to leptin-associated nephropathy in metabolic syndrome. In addition, our findings could have implications for lupus nephritis, an inflammatory kidney disease associated with the autoimmune disease systemic lupus erythematosus. Patients with this complication show activated JAK-STAT and elevated leptin, regulated by sterol regulatory element binding transcription factor 1 (SREBF1), which is involved in lipogenesis (*Hao et al., 2013*). Already, phase 2/3 clinical trials are underway to study the efficacy of JAK inhibitors in treating lupus nephritis (*Huo et al., 2023*). Our data indicate a possible second pathway could contribute, by direct leptin stimulation of JAK-STAT caused by the dysregulated fat body, warranting further investigation.

Altogether, our study expands our understanding of chronic kidney disease associated with super-fluous fat intake and provides new avenues for therapeutic strategies.

# Materials and methods

## Key resources table

| Reagent type (species) or resource | Designation | Source or reference | Identifiers | Additional information |
|---|---|---|---|---|
| Antibody | Chicken polyclonal anti-GFP | Abcam | Cat. ab13970; RRID:AB_300798 | IF(1:1000) |
| Antibody | Mouse monoclonal anti-Pyd | Developmental Studies Hybridoma Bank (DSHB) | RRID:AB_2618043 | IF(1:100) |
| Antibody | Goat anti-mouse Alexa Fluor 488 | Invitrogen | Cat. A11029; RRID:AB_2534088 | IF(1:500) |
| Antibody | Goat anti-chicken Alexa Fluor 488 | Invitrogen | Cat. A11039; AB_2534096 | IF(1:500) |
| Chemical compound, drug | Methotrexate | Sigma-Aldrich | Cas. 06563 | Methotrexate treatment |
| Other | DAPI | Thermo Fisher Scientific | Cat. D1306 | Immunochemistry |
| Other | 10 kD Texas Red-dextran | Thermo Fisher Scientific | Cas. D1828 | FITC-albumin and 10 kD dextran uptake assays |
| Other | FITC-albumin solution | Sigma | Cas. A9771 | FITC-albumin and 10 kD dextran uptake assays |
| Genetic reagent (*D. melanogaster*) | *Drosophila melanogaster*: $w^{1118}$ | Bloomington *Drosophila* Stock Center (BDSC) | RRID:BDSC_3605 | |
| Genetic reagent (*D. melanogaster*) | *Drosophila melanogaster*: *Dot*-Gal4 | BDSC | RRID:BDSC_67608 | |
| Genetic reagent (*D. melanogaster*) | *Drosophila melanogaster*: *ppl*-Gal4 | BDSC | RRID:BDSC_58768 | |
| Genetic reagent (*D. melanogaster*) | *Drosophila melanogaster*: *tub*-Gal80ts | BDSC | RRID:BDSC_7017 | |
| Genetic reagent (*D. melanogaster*) | *Drosophila melanogaster*: *10XStat92E*-GFP | BDSC | RRID:BDSC_26198 | |
| Genetic reagent (*D. melanogaster*) | *Drosophila melanogaster*: *Stat92E-IR_#2* | BDSC | RRID:BDSC_33637 | |
| Genetic reagent (*D. melanogaster*) | *Drosophila melanogaster*: *Socs36E*-IR | BDSC | RRID:BDSC_35036 | |
| Genetic reagent (*D. melanogaster*) | *Drosophila melanogaster*: *Stat92E*-IR | Vienna *Drosophila* Resource Center (VDRC) | VDRC_106980 | |
| Genetic reagent (*D. melanogaster*) | *Drosophila melanogaster*: *sns-mRuby3* | **Delaney et al., 2024** | | |
| Genetic reagent (*D. melanogaster*) | *Drosophila melanogaster*: hs-Flp$^{122}$; UAS-Flp$^{JD1}$/CyO, Act-GFP$^{JMR1}$; Act5C>CD2>Gal4$^S$, UAS-mCD8-GFP$^{LL6}$/TM6b | **Zhao et al., 2015** | | |
| Genetic reagent (*D. melanogaster*) | *Drosophila melanogaster*: UAS-hop.Tum | **Harrison et al., 1995** | | |
| Genetic reagent (*D. melanogaster*) | *Drosophila melanogaster*: UAS-upd2:GFP | **Hombría et al., 2005** | | |
| Software, algorithm | FIJI (ImageJ) | **Schneider et al., 2012**; https://imagej.net/Fiji/Downloads | Fiji-macOS | RRID:SCR_003070 |
| Software, algorithm | Adobe Illustrator | https://www.adobe.com/ | Adobe Illustrator 2022 | RRID:SCR_010279 |
| Software, algorithm | GraphPad Prism | https://www.graphpad.com/scientific-software/prism/ | GraphPad Prism 9 | RRID:SCR_002798 |

## *Drosophila* husbandry

Fly lines were reared on a normal fat diet (NFD; Nutri-Fly German formula; Genesee Scientific, San Diego, CA) or a high-fat diet (HFD; NFD supplemented with 14% coconut oil), under standard conditions (25°C, 60% humidity, 12 hr:12 hr dark:light cycle), unless otherwise stated.

Drosophila stocks $w^{1118}$ (BDSC_3605), *Dot*-Gal4 (BDSC_67608), *ppl*-Gal4 (BDSC_58768), *tub*-Gal80ts (BDSC_7017), *10XStat92E*-GFP (BDSC_26198), UAS-*Stat92E*-RNAi_#2 (BDSC_33637), and UAS-*Socs36E*-RNAi (BDSC_35036) were obtained from the Bloomington *Drosophila* Stock Center (BDSC). UAS-*Stat92E*-RNAi (VDRC_106980) was obtained from the Vienna *Drosophila* Resource Center (VDRC). UAS-*hop.Tum* was kindly provided by Prof. Norbert Perrimon (Harvard Medical School, Boston, MA; Howard Hughes Medical Institute, Boston, MA). UAS-*upd2:GFP* has been previously

described (*Hombría et al., 2005*) and was kindly provided by Prof. Ylva Engström (Stockholm University, Stockholm, Sweden). The Flp-out line *hs-Flp*[122]; UAS-*Flp*[JD1]/CyO, *Act-GFP*[JMR1]; *Act5C>CD2>Gal4*[S], UAS-*mCD8-GFP*[LL6]/TM6b was generated previously (*Zhao et al., 2015*).

## FITC-albumin and 10 kD dextran uptake assays

Nephrocyte functional assays were performed ex vivo at room temperature, following a previously described method (*Wen et al., 2020*) with minor changes. *Drosophila* females were dissected in Schneider's *Drosophila* Medium (Thermo Fisher Scientific, MA), then incubated in a 10 kD Texas Red-dextran solution (0.05 mg/mL; D1828, Thermo Fisher Scientific, MA) in Schneider's *Drosophila* Medium (Thermo Fisher Scientific, MA) for 20 min, or a FITC-albumin solution (10 mM; Sigma, A9771) in Schneider's *Drosophila* Medium (Thermo Fisher Scientific, MA) for 5 min. The specimens were washed in artificial hemolymph twice, followed by fixation in 4% paraformaldehyde (PFA) for 60 min. Then the fixed specimens were washed thrice for 5 min in 1x phosphate buffered saline (1xPBS; pH 7.4) and mounted using Vectashield mounting medium (H-1000, Vector Laboratories, CA). FITC-albumin and 10 kD dextran specimens were imaged using a ZEISS LSM900 confocal microscope with ZEISS Zen acquisition software (blue edition; version 3.0) using a 20× Plan-Apochromat 0.8 N.A. air objective (ZEISS, Oberkochen, Germany). For quantitative comparison of fluorescence intensities, settings for the control condition were chosen to avoid oversaturation (using Range Indicator in ZEN blue; limiting the observed red dots to avoid oversaturation), then applied across the images for all samples/conditions within the assay. The fluorescence intensity of FITC-albumin and 10 kD dextran nephrocytes was determined using Fiji software (Image J; *Schneider et al., 2012*, version 2.9.0/1.53t; National Institutes of Health, Bethesda). For quantitation, for each genotype, the relative fluorescence intensity of 30 nephrocytes from 6 female flies (five nephrocytes/fly) was analyzed.

## Immunochemistry

Immunostaining was performed as previously reported (*Zhao et al., 2022*) with minor changes. Flies were briefly rinsed in 95% ethanol and dissected in 1xPBS at room temperature. Specimens were incubated in primary antibodies overnight at 4°C. The incubations in secondary antibodies were performed either overnight at 4°C or for 2 hr at room temperature. The following antibodies were used: chicken anti-GFP (1:1,000; ab13970, RRID:AB_300798, Abcam, Cambridge, UK), mouse monoclonal anti-Pyd (1:100; RRID:AB_2618043, Developmental Studies Hybridoma Bank, IA), goat anti-mouse Alexa Fluor 488 (1:500; A11029, RRID:AB_2534088, Invitrogen, Eugene, OR), and goat anti-chicken Alexa Fluor 488 (1:500; A11039, AB_2534096, Invitrogen, Eugene, OR). DAPI (0.5 mg/ml in PBST (0.2% Triton X-100 in 1x PBS); D1306, Thermo Fisher Scientific, MA) was used to visualize the nuclei. The nephrocytes were imaged using a ZEISS LSM900 confocal microscope (under Airyscan mode for Pyd images) with ZEISS ZEN acquisition software (blue edition; version 3.0) and a 63x Plan-Apochromat 1.4 N.A. oil objective (ZEISS, Oberkochen, Germany). For quantitative comparison of fluorescence intensities, settings for the control condition were chosen to avoid oversaturation (using Range Indicator in ZEN blue; limiting the observed red dots to avoid oversaturation), then applied across the images for all samples/conditions within the assay. Image J (*Schneider et al., 2012*) was used for image processing (version 2.9.0/1.53t; National Institutes of Health, Bethesda, MD). Typically, six flies were imaged per condition, and representative images for each are displayed in the figures.

## Transmission electron microscopy (TEM)

TEM was performed using standardized procedures. In brief, female adults (7 days old) were dissected in Schneider's *Drosophila* Medium (Thermo Fisher Scientific, MA). The heart tube and the attached nephrocytes were dissected, removed, and fixed using Sorensen phosphate buffer (2% PFA, 2.5% EM grade glutaraldehyde, 2 mM CaCl$_2$, 0.1 M NaOH; provided by the Electron Microscopy Core Imaging Facility at the Center for Innovative Biomedical Resources (CIBR), University of Maryland School of Medicine, MD). The processed specimens were imaged using a Philips CM100 TEM, carried out at the Electron Microscopy Core Imaging Facility at the Center for Innovative Biomedical Resources (CIBR) (University of Maryland School of Medicine, MD). The LC-LC distances were measured using the Straight Line tool in Fiji (Image J *Schneider et al., 2012*, version 2.9.0/1.53t; National Institutes of Health, Bethesda, MD) to connect two adjacent lacuna channels (LCs); a straight line was drawn from the middle of the first LC to the middle of the sixth LC, covering five LC-LC intervals on a TEM

image, then the measure function was used to obtain the distance value. In total, 60 LC-LC distances were measured per condition in 10 nephrocytes from six 7-day-old female flies. The presence of a dark electron-dense structure in the vacuoles was manually determined and counted in images obtained from 12 nephrocytes for NFD and 29 nephrocytes from HFD from six 7-day-old female flies.

### Tissue mosaic analysis

Flp-out clone (*Struhl and Basler, 1993*) induction was performed as previously described (*Duan et al., 2020*). In brief, female virgins of *hs-Flp*[122]; UAS-*Flp*[JD1]/CyO, *Act-GFP*[JMR1]; *Act5C>CD2>Gal4*[S], UAS-*mCD8-GFP*[LL6]/TM6b (*Zhao et al., 2015*) were crossed with UAS-*hop.Tum* males. The embryos were collected for 24 hr. First instar larvae (24 hr after embryo collection) received a 10 min heat shock in a 37°C water bath to induce the mosaic clones. After the heat shock, the larvae were maintained at 25°C. One-day-old female adults were subjected to the 10 kD dextran functional assay (described above). The GFP-positive nephrocyte clones and their neighboring nephrocytes were analyzed.

### Methotrexate treatment

Methotrexate (06563, Sigma-Aldrich, MO) was dissolved in DMSO to make a 10 mM stock solution. Flies were dissected as described for the nephrocyte functional assays. The dorsal cuticle (with nephrocytes) was transferred to methotrexate solution (10 µM) in Schneider's *Drosophila* Medium (Thermo Fisher Scientific, MA) and incubated at room temperature for 60 min. The samples incubated in Schneider's Medium supplemented with DMSO vehicle were used as a control. The specimens were rinsed with Schneider's *Drosophila* Medium (Thermo Fisher Scientific, MA), then subjected to a functional assay or fixed in 4% PFA for immunochemistry.

### Nie Red staining

Newly eclosed *w*[1118] flies were fed either a high-fat diet (HFD) or a control diet (CD) for 7 days before staining for lipid droplets using Nile Red. For Nile Red staining, nephrocytes were dissected in 1× PBS and fixed in 4% paraformaldehyde for 1 hr at room temperature. The samples were then washed three times with 1× PBS for 5 min each, followed by incubation with 0.1 µg/mL Nile Red (HY-D0718, MedChemExpress) for 10 min at room temperature. Afterward, the samples were rinsed three more times with 1x PBS for 5 min each.

### Data analyses and figure preparation

Image J (*Schneider et al., 2012*) (version .9.0/1.53t; National Institutes of Health, Bethesda, MD) was used to process the raw data of confocal images and to measure the relative fluorescence intensity. The data sets were tested for normality using the Shapiro-Wilk test and plotted using GraphPad Prism 9 software (version 9.5.1). Normally distributed data were analyzed by the two-tailed Student's t-test, or by two-way ANOVA with Sidak correction. $P<0.05$ was considered significant. The figures were arranged using Adobe Illustrator software (version 2022 26.2.1). Box plots show the median (center line), interquartile range (25th–75th percentiles; box), and whiskers extending to the minimum and maximum values.

## Acknowledgements

We thank the Bloomington *Drosophila* Stock Center (BDSC) based at Indiana University (Bloomington, IN), the Vienna *Drosophila* Resource Center (VDRC) based at Vienna BioCenter (Vienna, Austria), Prof. Norbert Perrimon (Harvard Medical School, Boston, MA; Howard Hughes Medical Institute, Boston, MA), and Prof. Ylva Engström (Stockholm University, Stockholm, Sweden) for sharing *Drosophila* stocks; and the Developmental Studies Hybridoma Bank (DSHB) based at the University of Iowa (Iowa City, IA) for providing the antibodies. Finally, we thank the Electron Microscopy Core Imaging Facility at the Center for Innovative Biomedical Resources (CIBR) (University of Maryland School of Medicine, MD, USA) for their support in TEM image acquisition.

# Additional information

### Funding

| Funder | Grant reference number | Author |
|---|---|---|
| National Institute of Diabetes and Digestive and Kidney Diseases | R01-DK098410 | Zhe Han |
| National Institute of Diabetes and Digestive and Kidney Diseases | R01-DK140937 | Zhe Han |

The funders had no role in study design, data collection and interpretation, or the decision to submit the work for publication.

### Author contributions

Yunpo Zhao, Conceptualization, Investigation, Methodology, Writing – original draft, Writing – review and editing; Jianli Duan, Conceptualization, Investigation, Methodology, Writing – original draft; Hannah Seah, Investigation; Joyce van de Leemput, Writing – original draft, Writing – review and editing; Zhe Han, Conceptualization, Resources, Formal analysis, Supervision, Funding acquisition, Writing – original draft, Project administration, Writing – review and editing

### Author ORCIDs

Yunpo Zhao ⓘ https://orcid.org/0000-0002-7942-3406
Zhe Han ⓘ https://orcid.org/0000-0002-5177-7798

Reviewer #1 (Public review): https://doi.org/10.7554/eLife.96987.3.sa1
Reviewer #2 (Public review): https://doi.org/10.7554/eLife.96987.3.sa2
Author response https://doi.org/10.7554/eLife.96987.3.sa3

# Additional files

### Supplementary files

MDAR checklist

Source data 1. Individual values for each condition and genotype shown in the plots.

### Data availability

All relevant data can be found within the article and its supplementary information. Requests for resources and reagents related to this manuscript should be directed to and will be fulfilled by the lead contact, Dr. Zhe Han (zhan@som.umaryland.edu).

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
