## [Editor Report · eLife Assessment]

This study presents **important** new insights linking obesity to kidney disease using a Drosophila model. A series of **compelling** experiments demonstrate that a high-fat diet induces excretion of a leptin-like JAK-STAT ligand from fat body, driving the adipose-nephrocyte axis through activated JAK-STAT signaling and subsequently causing a functional defect in nephrocytes. The approach using combination of genetic tools and pharmacological intervention is **solid** and confirms the mechanistic link, together with phenotypic analysis that further supports the authors conclusions.

---

## [Referee Report · Reviewer #1 (Public review)]

Summary:

Zhao and colleagues employ Drosophila nephrocytes as a model to investigate the effects of a high-fat diet on these podocyte-like cells. Through a highly focused analysis, they initially confirm previous research in their hands demonstrating impaired nephrocyte function and move on to observe the mislocalization of a slit diaphragm-associated protein (pyd) and a knock-in into the locus of the Drosophila nephrin (sns). Employing another reporter construct, they identify activation of the JAK/STAT signaling pathway in nephrocytes. Subsequently, the authors demonstrate the involvement of this pathway in nephrocyte function from multiple angles, using a gain-of-function construct, silencing of an inhibitor, and ectopic overexpression of a ligand. Silencing the effector Stat92E via RNAi or inhibiting JAK/STAT with Methotrexate effectively restored impaired nephrocyte function and slit diaphragm architecture induced by a high-fat diet, while showing no impact under normal dietary conditions.

Strengths:

The findings establish a link between JAK/STAT activity and the impact of a high-fat diet on nephrocytes. This nicely underscores the importance of organ crosstalk for nephrocytes and supports a potential role for JAK/STAT in diabetic nephropathy, as previously suggested by other models.

Weaknesses:

While the analysis provides valuable insights, it appears somewhat over-reliant on tracer uptake in certain instances. Clinical inferences based on a Drosophila model should be interpreted with caution.

---

## [Referee Report · Reviewer #2 (Public review)]

Summary:

In their manuscript, Zhao et al. describe a link between JAK-STAT pathway activation in nephrocytes upon a high-fat diet. Nephrocytes are the homologs to mammalian podocytes, and it has been previously shown that metabolic syndrome and obesity is associated with worse outcomes for chronic kidney disease. A study from 2021 (Lubojemska et al.) could already confirm a severe nephrocyte phenotype upon feeding Drosophila a high fat diet and also linking lipid overflow by expressing adipose triglyceride lipase in the fat body to nephrocyte dysfunction. In this study, the authors identified a second pathway and mechanism, how lipid dysregulation impact on nephrocyte function. In detail, they show an activation of JAK-STAT signaling in nephrocytes upon feeding a high-fat diet, which was induced by Upd2 expression (a leptin-like hormone) in the fat body, the adipose tissue in Drosophila. Further, they could show genetic and pharmacological interventions can reduce JAK-STAT activation and thereby prevent the nephrocyte phenotype in the high-fat diet model.

Strengths:

The strength of this study is the combination of genetic tools and pharmacological intervention to confirm a mechanistic link between the fat body/adipose tissue and nephrocytes. Inter-organ communication is crucial in the development of several diseases, but the underlying mechanisms are only poorly understood. Using Drosophila, it is possible to investigate several players of one pathway, here JAK-STAT. This was done, by investigating the functional role of Hop, Socs36E and Stat92E in nephrocytes and has also been combined with feeding a high-fat diet, to assess restoration of nephrocyte morphology and function by inhibiting JAK-STAT signaling. Adding a translational approach was done by inhibiting JAK-STAT signaling with methotrexate, which also resulted in attenuated nephrocyte dysfunction. Expression of the leptin-like hormone upd2 in the fat body is a good approach to study inter-organ communication and the impact of other organs/tissue on nephrocyte function and expands their findings from nephrocyte function towards whole animal physiology.

Weaknesses:

Although the general findings of this study are of great interest, the number of flies investigated for the majority of the experiments is very low (6 flies). Also it is not clear whether the 6 flies used are from independent experiments to exclude differences in food/diet.

---

## [Author Response]

The following is the authors’ response to the original reviews

**Public Reviews:**

**Reviewer #1 (Public Review):**
Summary:Zhao and colleagues employ Drosophila nephrocytes as a model to investigate the effects of a high-fat diet on these podocyte-like cells. Through a highly focused analysis, they initially confirm previous research in their hands demonstrating impaired nephrocyte function and move on to observe the mislocalization of a slit diaphragmassociated protein (pyd). Employing a reporter construct, they identify the activation of the JAK/STAT signaling pathway in nephrocytes. Subsequently, the authors demonstrate the involvement of this pathway in nephrocyte function from multiple angles, using a gain-of-function construct, silencing of an inhibitor, and ectopic overexpression of a ligand. Silencing the effector Stat92E via RNAi or inhibiting JAK/ STAT with Methotrexate effectively restored impaired nephrocyte function induced by a high-fat diet, while showing no impact under normal dietary conditions.Strengths:The findings establish a link between JAK/STAT activity and the impact of a high-fat diet on nephrocytes. This nicely underscores the importance of organ crosstalk for nephrocytes and supports a potential role for JAK/STAT in diabetic nephropathy, as previously suggested by other models.Weaknesses:The analysis is overly reliant on tracer endocytosis and single lines. Immunofluorescence of slit diaphragm proteins would provide a more specific assessment of the phenotypes.

We thank the reviewer for the positive comments and pointing out that slit diaphragm markers would provide a more specific assessment of the phenotypes. In our revised manuscript, we used Sns-mRuby3, in which mRuby3 was tagged endogenously at the C-terminal of Sns (PMID: 39195240 and PMID: 39431457), to show the slit diaphragm pattern.

**Reviewer #2 (Public Review):**
Summary:In their manuscript, Zhao et al. describe a link between JAK-STAT pathway activation in nephrocytes on a high-fat diet. Nephrocytes are the homologs to mammalian podocytes and it has been previously shown, that metabolic syndrome and obesity are associated with worse outcomes for chronic kidney disease. A study from 2021 (Lubojemska et al.) could already confirm a severe nephrocyte phenotype upon feeding Drosophila a high-fat diet and also linking lipid overflow by expressing adipose triglyceride lipase in the fat body to nephrocyte dysfunction. In this study, the authors identified a second pathway and mechanism, how lipid dysregulation impact on nephrocyte function. In detail, they show activation of JAK-STAT signaling in nephrocytes upon feeding them a high-fat diet, which was induced by Upd2 expression (a leptin-like hormone) in the fat body, and the adipose tissue in Drosophila. Further, they could show genetic and pharmacological interventions can reduce JAK-STAT activation and thereby prevent the nephrocyte phenotype in the high-fat diet model.Strengths:The strength of this study is the combination of genetic tools and pharmacological intervention to confirm a mechanistic link between the fat body/adipose tissue and nephrocytes. Inter-organ communication is crucial in the development of several diseases, but the underlying mechanisms are only poorly understood. Using Drosophila, it is possible to investigate several players of one pathway, here JAK-STAT. This was done, by investigating the functional role of Hop, Socs36E, and Stat92E in nephrocytes and has also been combined with feeding a high-fat diet, to assess restoration of nephrocyte function by inhibiting JAK-STAT signaling. Adding a translational approach was done by inhibiting JAK-STAT signaling with methotrexate, which also resulted in attenuated nephrocyte dysfunction. Expression of the leptin-like hormone upd2 in the fat body is a good approach to studying inter-organ communication and the impact of other organs/tissue on nephrocyte function and expands their findings from nephrocyte function towards whole animal physiology.Weaknesses:Although the general findings of this study are of great interest, there are some weaknesses in the study, which should be addressed. Overall, the number of flies investigated for the majority of the experiments is very low (6 flies) and it is not clear whether the flies used, are from independent experiments to exclude problems with food/diet. For the analysis, the mean values of flies should be calculated, as one fly can be considered a biological replicate, but not all individual cells. By increasing the number of flies investigated, statistical analysis will become more solid. In addition, the morphological assessment is rather preliminary, by only using a Pyd antibody. Duf or Sns should be visualized as well, also the investigation of the different transgenic fly strains studying the importance of JAK-STAT signaling in nephrocytes needs to include a morphological assessment. Moreover, the expected effect of feeding a high-fat diet on nephrocytes needs to be shown (e.g. by lipid droplet formation) and whether upd2 is actually increased here should also be assessed. The time points of assessment vary between 1, 3, and 7 days and should be consistent throughout the study or the authors should describe why they use different time points.

We thank the reviewer for the comments and suggestions. HFD causes enlarged crop (Liao et al, 2021, PMID: 33171202) and accumulation of lipid droplets in the intestine. To exclude the problems with different batches of food/diet, we checked crop and the intestine during the sample preparation as indications of food consistency.

We followed the suggestion to take the mean values of flies in the data analysis, one was considered a biological replicate in the revised version. We added in another slit diaphragm protein reporter Sns-mRuby3, in which mRuby3 fluorescent protein was tagged at the C-terminal of endogenous Sns. This reporter was used to show the effect of HFD on slit diaphragm protein, manipulation of Jak/Stat pathway (*ppl-Gal4>upd2* and *dot-Gal4>UAS-Stat92E-RNAi*), and drug treatment.

Lubojemska et al 2021 (PMID: 33945525) showed that HFD leads to lipid droplet accumulation in larval nephrocytes. Following the reviewer’s suggestion, we stained the adult nephrocytes with Nile red and found lipid droplet formation caused by HFD, verifying the HFD effects on lipid droplet accumulation.

Regarding the timepoints, the newly eclosed flies (1-day old) were treated for 7 days (transferred to fresh diet or shifted from 18 to 29 °C for 7 days to induce target gene expression). Thus, the flies were 7 days old. In the revised manuscript, we changed “1-day-old females” to “7-day-old females” in the figure legend. The exception was Figure 4 panel G and H, we used Day 3 for the *UAS-hop.Tum* overexpression in the flp-out clones, which is different from the HFD approach (Day 7). This is because Hop.Tum is a strong gain of function mutation. *UAS-hop.Tum* overexpression in the eye imaginal disc leads to apoptosis via up-regulating a proapoptotic gene *hid* (Bhawana Maurya et al, 2021, PMID: 33824299). Thus, we used Day 3 for this experiment.

**Recommendations for the authors:**

**Reviewer #1 (Recommendations For The Authors):**
There are relevant issues, that should be addressed:Major:- The analysis of JAK/STAT signaling in nephrocytes is limited to nephrocyte function, despite the nice slit diaphragm phenotype shown in Figure 2A. What happens to the slit diaphragm in the other genotypes, the rescue settings in particular? Immunofluorescence of Pyd should be explored for all conditions to evaluate proper phenocopy. Tracer endocytosis is much less specific.

We thank the reviewer for the suggestion. We made a transgenic line Sns-mRuby3, in which mRuby3 was tagged to the endogenous Sns C-terminal. It has been used as a slit diaphragm reporter (PMID: 39195240 and PMID: 39431457). Apart from the tracer assays, we used Sns-mRuby3 reporter and/or Pyd staining to visualize the changes in slit-diaphragm structures.

- The interventions are restricted to single RNAi lines and reporters, raising concerns about specificity/potential off-targets. Additional lines should be tested for verification.

Different versions of RNAi lines are available for targeting fly genes. For UAS-*Socs36E*-RNAi, we chose the one that was generated with a short hairpin, which is known to restrict the off-target effects (Ni et al, 2011, PMID: 21460824). For UAS-*Stat92E*-RNAi, we added in an independent RNAi line (Figure 6 - figure supplement 1 and 2).

Minor:- In Figure 2C, the image of HFD shows a section that cuts through the surface at a shallower angle, making everything appear blurry. This image should be replaced.

We replaced Figure 2C (the image of HFD) with another one.

- What is the relevance (if any) of reduced electrodense vacuoles with a high-fat diet? An effect on endocytic trafficking/endosome architecture remains unexplored.

Lubojemska et al (PMID: 33945525) studied the endocytic trafficking/endosome architecture of the larval nephrocytes and found that HFD impaired the endocytosis. We studied the adult pericardial nephrocytes. It is very likely that the endocytic trafficking/endosome architecture is affected by HFD in the adult nephrocytes.

- How do the findings presented in this manuscript correlate with a similar study by Lubojemska et al.? At least the discussion should provide more evaluation of this aspect.

Lubojemska et al (PMID: 33945525) assayed the larval nephrocytes and found that a HFD leads to the ectopic accumulation of lipid droplets in the nephrocytes and decreased endocytosis. They further demonstrated that lipid droplet lipolysis and PGC1α counteracts the harmful effects of a HFD. We performed Nile red staining and verified the accumulation of lipid droplets in the adult pericardial nephrocytes upon HFD feeding, which agrees with Lubojemska discovery. We found that a HFD activates Jak/Stat pathway, which mediates the nephrocyte functional defects. A previous study showed that Stat1 has an inhibitory effect on PGC1α transcription (PMID: 26689548). Further study is needed to investigate the interaction between Jak/Stat pathway and PGC1α transcription. We added the information to the discussion.

- Please check spelling and grammar.
**Reviewer #2 (Recommendations For The Authors):**
(1) Which cells are investigated? Please state.

Pericardial nephrocytes were used in this study. The information was added to the result parts.

(2) Rephrase 'chronic kidney disease model'. Feeding for 7 days and assessment after 7 days cannot be considered chronic as flies can live more than 60 days.

Lubojemska et al (PMID: 33945525) fed the newly hatched larvae with a HFD and used the third instar larvae for the experiments. The term “chronic kidney disease” has been used in the reference PMID: 33945525. It takes about 4 days for fly larvae to develop from the first instar to the third instar. Thus, the animals were fed on the HFD for only 4 days. In this regard, feeding for seven days might be considered as chronic.

(3) Line 89: (Curran et al., 2014). with risk increasing risk as BMI increases (Hsu et al., 2006). Please correct this sentence.

We thank the reviewer for finding the error. In the revised version, the sentence was changed as “with increasing risk as BMI increases (Hsu et al., 2006)”.

(4) Figure 1: The authors should explain why they use FITC-Albumin and 10kDA dextran, what are the differences, and why are both used?

The tracers are different in size (70kD FITC-Albumin and 10kDA dextran). Both FITC-Albumin and 10kDA dextran have been used in previous publications (Zhao et al 2024, PMID: 39431457 and Weavers et al 2009, PMID: 18971929) to show that the nephrocytes can efficiently take up the tracers of different sizes.

(5) Figure 3: The JAK-STAT sensor was used on Day 1 to confirm activation of JAKSTAT signaling, which means a very fast response towards the HFD after 24hrs. How is the activation after 7 days? The nephrocyte assessment in Figures 1 and 2 is done at the later time point, how about earlier time points in HFD? One would expect an earlier phenotype as well if JAK-STAT signaling is causative.

In Figure 3C, newly eclosed flies (1-day old) were fed on a control diet or a HFD for 7 days. Thus, in the legend it shall be “7-day-old females”. Sorry for misleading. The caption was updated as “7-day-old females”.

(6) Figure 4H: I don't understand how many cells or flies are depicted and analysed? Are the dots one nephrocyte from 4 flies? If yes, the numbers need to be increased.

In figure 4H, we quantified 5 *UAS-hop.Tum* clones and 5 neighbor cells. We only found 5 clones from 4 flies. We didn’t quantify all the nephrocytes, since we compared the clone with its neighbor cell. To make it easier to follow, we changed the description as “n=5 clones and 5 neighbor cells”.

(7) Figure 4: Why are flies investigated at different ages? Day 1 vs Day 3? This should be consistent with the HFD approach and day 7. Or investigate the HFD at earlier time points as well.

In Figure 4, the newly eclosed flies (1-day old) were shifted from 18 to 29 °C for 7 days to induce target gene expression. Thus, the flies were 7-day old. In the revised manuscript, we changed “1-day-old females” to “7-day-old females” in the figure legend. We used Day 3 for the UAS-hop.Tum overexpression in the flp-out clones, which is different from the HFD approach (Day 7). This is because Hop.Tum is a strong gain of function mutation. *UAS-hop.Tum* overexpression in the eye imaginal disc leads to apoptosis via up-regulating a proapoptotic gene *hid* (Bhawana Maurya et al, 2021, PMID: 33824299). Thus, we used Day 3 for this experiment.

(8) Figure 5: Do the authors see upd2-GFP in the nephrocyte or at the nephrocyte? Is upd2 filtered to bind the JAK-STAT-receptor? They should show this, which is easy to do due to the GFP label.

We thank the reviewer for the suggestion. We looked into the nephrocyte from *ppl-Gal4>upd2-GFP* flies and found Upd2-GFP in the nephrocytes. We further showed that *ppl-Gal4* was not expressed in the nephrocytes, suggesting that Upd2-GFP is secreted from the fat body and transported to the nephrocytes. We stained the nephrocytes for Pyd and found compromised fingerprint pattern caused by Upd2-GFP expression in the fat body. The data was added to Figure 5 - figure supplement 1.

(9) Figure 5: What are the upd2 levels after day 1 and compared to HFD at day 7? In the Rajan et al manuscript, upd2 levels have been assessed by qPCR, this can be done here as well. Although there is a mechanistic link shown here, I think it would be interesting to test the upd2 levels at the different time points assessed.

In the Rajan et al manuscript, they showed that the expression of upd2 was up regulated by HFD. My previous work showed that HFD changes taste perception. We performed qPCR to determine the expression of upd2 and verified that upd2 was upregulated in HFD fed flies (Yunpo Zhao et al. 2023. PMID: 37934669). We included the reference in the revised version.

(10) Figure 6: Does a Socs36E overexpression e.g. with the Bloomington strain 91352 also rescue the HFD phenotype, by blocking JAK-STAT signaling?

We thank the reviewer for the suggestion. We tested the effect of Socs36E overexpression and observed that *UAS-Socs36E* can partially rescue HFD caused nephrocyte functional decline. The data was not included in the revised manuscript. Notably, apart from having an inhibitory effect on the Jak/Stat, Socs36E represses MAPK pathway (Amoyel et al, 2016, PMID: 26807580).

(11) Figure 7: What is the control for the methotrexate treatment? What is the solvent?

We used DMSO as the solvent for methotrexate and used it as the control for the methotrexate treatment. We added the following sentences to the method parts, “Methotrexate (06563, Sigma-Aldrich, MO) was dissolved in DMSO to make a 10mM stock solution”, and “The samples incubated in Schneider’s Medium supplemented with DMSO vehicle were used a control”.

(12) Why did the authors use Dot-Gal4 for the Socs36E knockdown and Dot-Gal4ts for the Stat92E knockdown?

We used Dot-Gal4ts and temperature shifting to restrict the Stat92E knockdown at adult stages.

(13) Supplementary Figure 1: Please add the individual data to the figure as done for all other figures.

We thank the reviewer for this comment. The figure individual data was added according to the suggestion.